**Effect of the presence of plateau pikas on the ecosystem services of alpine meadows**
**Yingying Chen[1], Huan Yang[1], Gensheng Bao[2], Xiaopan Pang[1], Zhenggang Guo[1]**
[1]Engineering Research Center of Grassland Industry, Ministry of Education; Key Laboratory
of Grassland Livestock Industry Innovation, Ministry of Agriculture and Rural Affairs;
College of Pastoral Agriculture Science and Technology, Lanzhou University, Lanzhou,
730020, P. R. China
[2]Academy of Animal and Veterinary Sciences, Qinghai University (Qinghai Academy of
Animal and Veterinary Sciences), Xining, China
**Correspondence:** Zhenggang Guo (guozhg@lzu.edu.cn)

**Abstract**

The activity of small mammalian herbivores influences grassland ecosystem services in arid and semi-arid regions. Plateau pika (*Ochotona curzoniae*) was considered as a focal organism to investigate the effect of small mammalian herbivores on meadow ecosystem services in alpine regions. In this study, a home-range scale was used to measure the forage available to livestock, water conservation, carbon sequestration, and soil nutrient maintenance (total nitrogen, phosphorus, and potassium) in the topsoil layer; and a quadrat scale was used to assess the biodiversity conservation of alpine meadows. This study showed that the forage available to livestock and water conservation were 19 % and 16 % lower in the presence of plateau pikas than in their absence, and biodiversity conservation, carbon sequestration, soil nitrogen, and phosphorus maintenance were 15 %, 29 %, 10 % and 8.9 % higher in the presence of plateau pikas than in their absence. In contrast, it had no impact on soil potassium maintenance of meadow ecosystems in alpine regions. The forage available to livestock, biodiversity conservation, and soil nutrient maintenance of meadow ecosystems in alpine regions had maximum values as the disturbance intensity of plateau pikas increased; the water conservation tended to decrease linearly with the increasing disturbance intensity of plateau pikas. These results present a pattern of plateau pikas influencing the ecosystem services of meadow ecosystems in alpine regions, enriching our understanding of the small mammalian herbivores in relation to and grassland ecosystem service.

**1 Introduction**

Grasslands provide multiple ecosystem services, mainly including provisioning services of food and water, regulating services of carbon sequestration and water conservation, supporting services of soil nutrient maintenance and biodiversity conservation, and cultural

services of landscapes and recreation tourism (Millennium Ecosystem Assessment, 2005;
Bai and Cotrufo, 2022; Buisson et al., 2022; Strömberg and Staver, 2022). These ecosystem
services sustain animal production, flora and fauna, and other human welfare (Costanza et al.,
1997; Zhang et al., 2018; Dong et al., 2020); however, they are affected by multiple biotic
factors, such as soil microbial communities (Van Eekeren et al., 2010), grazing by large
herbivores (Lu et al., 2017), and the presence of small herbivores (Delibes-Mateos et al.,
2011; Martínez-Estévez et al., 2013).
Small mammalian herbivores are common biotic factors (Davidson et al., 2012). These
herbivores usually create extensive disturbances on grassland vegetation and soil (Pang et al.,
2020a, 2020b) by developing burrow systems (Delibes-Mateos et al., 2008; Sun et al., 2015),
excreting feces and urine (Zhang et al., 2016), consuming plants (Eldridge and Myers, 2001;
Liu et al., 2017), clipping tall plants (Zhang et al., 2020), and producing bare soil patches
(Guo et al., 2012a, 2012b; Yu et al., 2017a, 2017b) or mounds (Yang et al., 2021). Previous
studies have shown that the presence of prairie dogs (*Cynomys ludovicianus*) can increase the
forage available to livestock, water conservation, carbon sequestration, and biodiversity
conservation of grassland ecosystems in arid regions (Ceballos et al., 1999, Martínez-Estévez
et al., 2013), whereas the presence of European rabbit (*Oryctolagus cuniculus*) can decrease
the forage available to livestock (Delibes-Mateos et al., 2008; Eldridge and Myers, 2001), and
increase the biodiversity conservation (Delibes-Mateos et al., 2008) and nitrogen maintenance
(Willott et al., 2000) of grassland ecosystems in semi-arid regions. In addition to grasslands in
arid and semi-arid regions, vast alpine meadows exist in high latitude and altitude regions
throughout the world (Zhang et al., 2018; Dong et al., 2020). However, how small
mammalian herbivores influence the ecosystem services in alpine meadows as much as they
do in arid and semi-arid regions has not been well documented.

The plateau pika (*Ochotona curzoniae*) is a common, small mammalian herbivore that

mainly lives in alpine meadows of the Qinghai-Tibetan Plateau (Smith and Foggin, 1999).
This small mammalian herbivore with an average weight of 150 g are diurnally active and
non-hibernating (Smith and Wang, 1991; Fan et al., 1999), and preferentially consume
dicotyledons (Zhao et al., 2013; Pang and Guo, 2017). Plateau pikas, a sexual monomorphism
(Dobson et al., 1998), often construct a family warren with numerous burrow entrances and
develop a complex burrow system with an average length and depth of 13 m and 30 cm (Fan
et al., 1999). This mammalian herbivore is social and philopatric (Dobson et al., 1998) and its
young offspring stay with its family during its birth year (Wang et al., 2020). Plateau pikas are
generally considered a pest in China (Harris, 2010; Pang and Guo, 2017) as they often
exacerbate the degradation of alpine meadows (Liu et al., 2013; Zhang et al., 2016). However,
some studies have argued that plateau pika is a key species in alpine meadow ecosystems
(Smith and Foggin, 1999; Delibes-Mateos et al., 2011). This disagreement has encouraged
professionals to re-evaluate the role of plateau pikas in alpine meadow ecosystems. Thus, the
effects of plateau pikas' presence on ecosystem services of alpine meadows allow insight into
the role of plateau pikas in alpine meadow ecosystems. Previous studies have demonstrated
that the presence of plateau pikas decreases (Liu et al., 2013) or has no significant effect on
plant biomass (Pang and Guo, 2017), increases (Liu et al., 2017; Pang and Guo, 2017) or
decreases (Sun et al., 2015) plant-species richness, and increases (Yu et al., 2017a; Pang et al.,
2020a, 2020b) or decreases (Sun et al., 2015) soil carbon and nutrients. In addition, previous
studies have shown that the disturbance intensity of plateau pikas affects plant-species
richness, and soil nutrient stocks of alpine meadows (Yu et al., 2017a; Pang and Guo, 2018).
These findings imply that plateau pikas may have an impact on the ecosystem services of
alpine meadows. Thus, further studies are needed to test whether the presence of plateau pikas
and its disturbance intensity influence the ecosystem services of alpine meadows, which can
enrich our understanding of the presence of small mammalian herbivores in relation to
grassland ecosystem services.
Since soil carbon and nutrients differ between vegetated and bare soil patches in the
home range (Yu et al., 2017b), Pang et al. (2020a; 2020b) proposed that the home-range scale
is a better proxy than the quadrat scale to estimate the complete effects of the presence of
plateau pikas on soil carbon and nutrient stocks. Although the provisioning, regulation,
support, and cultural services of alpine meadows can be estimated by multiple indicators
(Egoh et al., 2012; Brown et al., 2014), one or two can be used to verify whether the presence
and intensity of plateau pikas influence each ecosystem service. In previous studies, palatable
plant biomass for livestock has been used to evaluate the provisioning services
(Martínez-Estévez et al., 2013; Wen et al., 2013); soil-water storage and soil organic carbon
stock have been used to evaluate the regulating services (Wen et al., 2013; Li and Xie, 2015;
Tang et al., 2019;); and plant-species richness and soil total nutrient stocks can be used to
evaluate the supporting services (Wen et al., 2013). Notably, cultural services are particularly
related to the spatial scale, as many are perceived visually over distant views (Norton et al.,
2012). The plateau pika is territorial and its habitat use is patchy within a given area (Pang et
al., 2020a), which leads to mismatches between the spatial scale and cultural services (De
Groot et al., 2010). Therefore, the present study used ecological services of forage available to
livestock, water conservation, carbon sequestration and soil nutrient maintenance, and
biodiversity conservation to test how the presence of plateau pikas influences the ecosystem
services of alpine meadows across five sites. In this study, we hypothesized that (1) the
presence of plateau pikas leads to lower forage available to livestock because of lower
palatable plant biomass in the presence of small mammalian herbivores; (2) the presence of
plateau pikas leads to higher water conservation and carbon sequestration because small
mammalian herbivores can increase soil-water storage and carbon stocks; and (3) the
presence of plateau pikas leads to higher biodiversity conservation and soil nutrient
maintenance because small mammalian herbivores can increase plant-species richness and
soil nutrient stocks.
**2 Materials and methods**
**2.1 Study site descriptions**
Plateau pikas can live in various habitats with different soil types, topographies, and
microclimates on the Qinghai-Tibetan Plateau. To determine how the presence of plateau
pikas generally influences the ecosystem services of alpine meadows, five survey sites were
selected in Luqu (102°22′12″E, 34°15′51″N), Gangcha (100°26′26″E, 37°36′12″N), Haiyan
(100°54′33″E, 36°57′50″N), Qilian (100°34′48″E, 37°43′26″N), and Gonghe (99°47′11″E,
36°43′48″N) counties on the Qinghai-Tibetan Plateau. These five survey sites have a similar
typical plateau continental climate, with elevations ranging from 3194 m at the Gonghe
survey site to 3550 m at the Luqu survey site. Based on 5-year weather data, the mean annual
temperatures are 3.1, 0.9, 1.9, 2.2, 3.3 °C at Luqu, Gangcha, Haiyan, Qilian, and Gonghe,
respectively, of which the average temperature in warm season are 9.3, 8.3, 9.6, 10.3, 9.9 ℃
and in cold season are -3.1, -6.5, -5.8, -5.9, -3.4 ℃. The mean annual precipitation is 439.5,
258.9, 257.4, 257.0, and 239.8 mm, of which the warm season accounts for 83.4 %, 92.8 %,
89.3 %, 91.5 %, 91.4 % at Luqu, Gangcha, Haiyan, Qilian, and Gonghe, respectively.
According to the Chinese soil classification system (Gong, 2001), the soil type at each site is
alpine meadow soil, similar to Cambisol in the WRB soil classification system.
Animal husbandry is the dominant use of alpine meadows on the Qinghai-Tibetan
Plateau, and herders traditionally graze their livestock seasonally on cold and warm
grasslands. The survey sites in this study were all situated in cold grasslands, in which alpine
meadows were fenced from mid-April to September, and fences were opened to grazing yaks
from mid-October to early April (Zhang et al., 2020). All field data were collected in August
when the annual population of plateau pikas was the highest and reproduction had largely
ceased (Qu et al., 2013; Pang et al., 2020a, 2020b). In addition, the growing season for plants
is short on the Qinghai-Tibetan Plateau, and some plants don't turn green until July. Therefore,
sampling in August is optimal because August is good time to identify all plants and ensure
an accurate survey of the plant species. Notably, the small burrowing herbivore at each survey
site was only plateau pikas.
**2.2 Field survey design**
The adult dispersal of plateau pikas is a gradual process (Pang et al., 2020a), it is easy to
identify reference sites without plateau pikas, even though these sites might be potential as
suitable habitats. In this study, a home-range scale was used to calculate the forage available
to livestock, water conservation, carbon sequestration, and soil nutrient maintenance, and a
quadrat scale was used to calculate the biodiversity conservation.
A stratified random and paired design was used to select plots. The home range of the
plateau pika was approximately 1262.5 m$^2$ (Fan et al., 1999), and the plot size was 35 × 35 m,
which was similar to the average area of the plateau pika's home range. At each of the five
sites, this study first selected 10 plots where plateau pikas were present, or where active
burrow entrances were observed. The second plot was identified along the road when the first
plot with plateau pikas was selected. The distance between the two plots with plateau pikas
was more than 3 km, which ensured that plateau pikas of the same family would not appear in
two plots at the same time. Second, a paired adjacent plot without plateau pikas and active
burrow entrances was selected for each plot with plateau pikas. The plots without plateau
pikas were in any direction of plots with plateau pikas. The distance between each plot with
plateau pikas and its paired plot without plateau pikas ranged from 500 to 1000 m. If the
distance between each paired plot was too close, the plateau pikas could move between plots
with and without plateau pikas. To ensure that each plot with plateau pikas was paired with a
plot without plateau pikas, each paired plot shared the alpine meadow with same dominant
plant, with no obvious differences in soil type, topography, or microclimate. In total, there
were 10 paired plots at each site and 100 plots across five sites, including 50 with and 50
without plateau pikas. Each paired plot shared the same grazing intensity during the cold
season; however, 50 paired plots consisted of different yak grazing intensity, and this can
permit the general pattern relating to the effect of plateau pika disturbance on alpine meadow
ecosystem services.
**2.3 Field sampling**

Field surveys and sampling were conducted in early August 2020. First, the active

burrow entrance at each plot with plateau pikas was estimated by the "plugging tunnels

method," in which the burrow entrances were plugged with hay for 3 days (Sun et al., 2015),

and the number of plugs cleared by the plateau pikas to allow access to the meadow surface

was recorded (Guo et al., 2012a). The average number of burrow entrances with cleared plugs

after 3 days was taken as the density of active burrow entrances. For plots with plateau pikas,

the density of active burrow entrances was used as a proxy for the intensity of the disturbance

(Guo et al., 2012a; Sun et al., 2015). Second, this study was restricted to plateau pikas in

relation to the ecosystem services of alpine meadows. However, bare soil patches caused by

other factors (no plateau pikas) is simultaneously existed on the vegetated surface in the

presence/absence of plateau pikas. To actual quantify the effect of plateau pikas on ecosystem

services of alpine meadows, this study only measured the area of bare soil patches caused by

plateau pikas, although there exist multiple types of bare soil patches in alpine meadows. The

soil bare patches caused by plateau pikas is easily to identify because one soil bare patch

caused by plateau pikas is paired with a visible burrow entrance (Pang et al., 2021a). Other

soil bare patches are not paired with visible burrow entrance. The area of each bare soil patch

(created by plateau pikas) in the plot with plateau pikas was measured. Each bare soil patch

was identified as regular shape or irregular shape. If one bare soil patch was identified as

regular shape, such as rectangle, circle, trapezoid, etc; a ruler was used to measure its length,

width, height, diameter, upper and lower bottom, and these data was used to calculate the area

of that bare soil patch. If one bare soil patch was identified as irregular shape, this bare soil

patch was divided into several regular shapes; the areas of these regular shapes were

calculated, respectively; the area sum of these regular shapes form irregular bare soil patch
was considered as the area of that irregular bare soil patch (Han et al., 2011). Then, the sum of
all bare soil patch areas in each plot with plateau pikas was calculated and defined as the bare
soil area for that plot. Third, five vegetated quadrats (1 × 1 m) were placed on the vegetated
surface approximately 8 m apart along the shape of a W pattern in all plots (with or without
plateau pikas). In plot with plateau pikas, if quadrat was justly covered with the bare patches
caused by plateau pikas, the quadrat was slightly moved to avoid it; if quadrat was justly
covered with the bare patches caused by other factors, the quadrat was not moved. Fourth,
alpine meadows in plot with plateau pikas consisted of bare and vegetated surface, and a bare
soil patch was selected as a paired bare soil quadrat for each vegetated quadrat in the plot with
plateau pikas, and the distance between each paired bare soil quadrat and vegetated quadrat
was as short as possible (less than 1 m). Bare soil patch quadrat and each vegetated quadrat
were beneficial to accurately measure the soil nutrient, carbon concentrations and plant
biomass, which reflected the effect of plateau pikas' presence on ecosystem services in alpine
meadows by comparing the parameters between plots with and without plateau pikas at home
range scale. Thus, there were 5 paired quadrats, consisting of 5 vegetated and 5 bare soil
quadrats in each plot with plateau pikas. Additionally, there were 5 vegetated quadrats in each
plot without plateau pikas, since this study focused on bare soil patches induced by plateau
pikas.
In each vegetated quadrat of the plot with or without plateau pikas, all vascular plant
species were identified, and the number of plant species were recorded as plant-species
richness. Then, all plants rooted in a quadrat were harvested into palatable and unpalatable
plants, in which palatable plants or unpalatable plants were for yak and Tibetan sheep (Pang
and Guo, 2017). Finally, all palatable plant samples were placed in envelopes and transported
to the laboratory.

Generally, most burrows derived from the activities of plateau pikas are less than 20 cm

in depth (Yu et al., 2017b), although a few burrows extend to depths of 60 cm (Fan et al.,
1999). In addition, the majority of plant roots in alpine meadows of the Qinghai-Tibetan
Plateau are in the top 20 cm of the soil. The soil samples were collected at a depth of 20 cm.
Soil samples were collected from vegetated and bare soil quadrats for each plot with plateau
pikas and vegetated quadrats for each plot without plateau pikas. Before collecting the soil
samples, plants and litter were removed from the soil surface. First, a 5 cm diameter soil
auger was used to collect soil samples, which were used to measure soil organic carbon and
soil nutrient concentrations (total nitrogen, phosphorus, and potassium). Second, soil profiles
in each quadrat were obtained using a spade, and a stainless-steel cutting ring (with a volume
of 100 cm$^3$) was used to collect soil cores to determine soil bulk density and soil water
content. Soil samples used to determine soil bulk density were packed into aluminum boxes
with recorded weights, and each aluminum box was numbered. The aluminum boxes
containing fresh soil were immediately weighed, recorded, stored at 4 °C, and then
transported to the laboratory. Thus, in this study, 10 soil samples were collected to analyze the
soil carbon, nitrogen, phosphorus, and potassium concentrations, and 10 soil samples were
obtained to determine the soil bulk density in each plot with plateau pikas. Because this study
is constricted with bare soil patch caused by plateau pikas, bare soil quadrats was not
considered in plot without plateau pikas; therefore, 5 soil samples were used to determine the
soil carbon, nitrogen, phosphorus, and potassium concentrations, and 5 samples were obtained
for the analysis of soil bulk density in each plot without plateau pikas. 5 soil samples in each
plot were individually measured. The average value of five soil samples in one plot was
considered as the representative data of that plot.
**2.4 Analysis of samples**
In the laboratory, palatable plant samples were dried in an oven at 80 °C for 48 h and
weighed. The soil samples used to measure soil bulk density and soil-water content were
dried to a constant weight at 105±2 °C, and the aluminum boxes with dry soil were weighed
and the values were recorded. The soil samples used to measure soil organic carbon, total
nitrogen, phosphorus, and potassium concentrations were air-dried, gravel and roots were
manually picked out, and the proportion of gravel larger than 2.0 mm in the soil sample was
determined by passing through a 2.0 mm sieve. Finally, soil samples were sieved at 0.15 mm
to analyze soil organic carbon, nitrogen, phosphorus, and potassium concentrations in the
laboratory. Soil organic carbon was measured using the dichromate heating-oxidation
(Naelson and Sommers, 1982). Soil total nitrogen concentration was measured using the
Kjeldahl procedure. Soil total phosphorus concentration was measured using the
Molybdenum blue colorimetric method. Soil total potassium concentration was measured
using flame photometry.
Soil bulk density, soil organic carbon, and nutrient concentrations (total nitrogen,
phosphorus, and potassium) were used to calculate the soil organic carbon, total nitrogen,
phosphorus, and potassium stocks. Soil bulk density and soil-water content were used to
calculate the soil-water storage (Jia et al., 2020).

**2.5 Calculations**

The bare soil area consisted of all bare soil patches, and the vegetated surface area was estimated from the plot areas minus the bare soil areas. This study only measured the area of bare soil patches caused by plateau pikas, although there were other kinds of bare soil patches in alpine meadows. Therefore, bare soil areas caused by plateau pikas were considered to be zero in each plot without plateau pikas, and the vegetated surface area was considered to be 100%.

The palatable plant biomass was calculated using the following equation:

$$GB=B_q \times \delta_{va} \tag{1}$$

where $GB$, $B_q$, and $\delta_{va}$ are the palatable plant biomass of the plot, palatable plant biomass on the quadrat scale (g m$^{-2}$), and vegetated surface area, respectively.

The plant-species richness in a quadrat ($1 \times 1$ m) was measured using the species number of each quadrat.

Soil-water storage was determined using the method recommended by Jia et al. (2020), and it was calculated by the following equation:

$$SWS_{pika}=(SWC_{BA} \times BD_{BA} \times T \times (1-\theta_{BA}) \times 0.01 \times BA)+(SWC_{VA} \times BD_{VA} \times T \times (1-\theta_{VA}) \times 0.01 \times VA) \tag{2}$$

Where $SWS_{pika}$, $SWC_{BA}$, $BD_{BA}$, and $\theta_{BA}$ were soil-water storage in a plot with plateau pikas, water content (g kg$^{-1}$), soil bulk density (g cm$^{-3}$) and soil fraction of gravel larger than 2 mm in  bare soil areas of plots with plateau pikas, respectively; $BA$ was the percentage of bare soil areas in plots with plateau pikas; $SWC_{VA}$, $BD_{VA}$, and $\theta_{VA}$ were water content (g kg$^{-1}$), soil bulk density (g cm$^{-3}$) and soil fraction of gravel larger than 2 mm in vegetated areas of a plot with plateau pikas, respectively; and $T$ was soil thickness (20 cm); $VA$ was the percentage

of vegetated surface area in plots with plateau pikas; $SWC_{BA}$ and $SWC_{VA}$ was measured by
oven drying method.
$$SWS_{no\ pika}=SWC_{no\ pika}\times BD_{no\ pika}\times T\times(1-\theta_{no\ pika})\times0.01\times100\% \quad\quad (3)$$
Where $SWS_{no\ pika}$, $SWC_{no\ pika}$, $BD_{no\ pika}$ and $\theta_{no\ pika}$ were soil-water storage in a plot without
plateau pikas, soil water content (g kg$^{-1}$), soil bulk density (g cm$^{-3}$) and soil fraction of gravel
larger than 2 mm in plots without plateau pikas, respectively; and $T$ is soil thickness (20 cm).
The soil organic carbon stock per plot was estimated using the method described by Pang
et al. (2020b), and it was calculated by following equation:
$$SOCS_{pika}=(SOC_{BA}\times BD_{BA}\times T\times(1-\theta_{BA})\times0.01\times BA)+(SOC_{VA}\times BD_{VA}\times T\times(1-\theta_{VA})\times0.01\times VA) \quad (4)$$
Where $SOCS_{pika}$ was soil organic carbon stock in a plot with plateau pikas (kg m$^{-2}$);
$SOC_{BA}$, $BD_{BA}$, and $\theta_{BA}$ were soil organic carbon concentration (g kg$^{-1}$), soil bulk density (g
cm$^{-3}$) and soil fraction of gravel larger than 2 mm in bare soil areas of plots with plateau pikas,
respectively; $BA$ was the percentage of bare soil areas in plots with plateau pikas; $SOC_{VA}$,
$BD_{VA}$, and $\theta_{VA}$ were organic carbon concentration (g kg$^{-1}$), soil bulk density (g cm$^{-3}$) and soil
fraction of gravel larger than 2 mm in vegetated areas of a plot with plateau pikas,
respectively; and T was soil thickness (20 cm); $VA$ was the percentage of vegetated surface
area in plots with plateau pikas.
$$SOCS_{no\ pika}=SOC_{no\ pika}\times BD_{no\ pika}\times T\times(1-\theta_{no\ pika})\times0.01\times100\% \quad\quad (5)$$
Where $SOCS_{no\ pika}$ was soil organic carbon stock in the plot without plateau pikas (kg
m$^{-2}$); and $SOC_{no\ pika}$, $BD_{no\ pika}$ and $\theta_{no\ pika}$ were soil organic carbon concentration (g kg$^{-1}$), soil
bulk density (g cm$^{-3}$) and soil fraction of gravel larger than 2 mm in plots without plateau
pikas, respectively; and $T$ was soil thickness (20 cm).
The soil total nitrogen, phosphorus, and potassium stocks per plot were quantified using
the method described by Pang et al. (2020a), and it was calculated by the following
equation:
$$SNSi_{pika}=(SNi_{BA} \times BD_{BA} \times T \times (1-\theta_{BA}) \times 0.01 \times BA)+(SNi_{VA} \times BD_{VA} \times T \times (1-\theta_{VA}) \times 0.01 \times VA) \quad (6)$$
Where $SNSi_{pika}$ was soil total nitrogen, phosphorus, potassium stock in plot with plateau
pikas (kg m$^{-2}$), and $SNi_{BA}$, $BD_{BA}$, and $\theta_{BA}$ were soil nutrient concentration (g kg$^{-1}$), soil bulk
density (g cm$^{-3}$) and soil fraction of gravel larger than 2 mm in bare soil area of plots with
plateau pikas, respectively; $BA$ was the percentage of bare soil areas in plots with plateau
pikas; $SNi_{VA}$, $BD_{VA}$, and $\theta_{VA}$ were soil nutrient concentration (g kg$^{-1}$), soil bulk density (g cm$^{-3}$)
and soil fraction of gravel larger than 2 mm in vegetated areas of a plot with plateau pikas,
respectively; and $T$ was soil thickness (20 cm); $VA$ was the percentage of vegetated surface
area in plots with plateau pikas.
$$SNSi_{no\ pika}=SNi_{no\ pika} \times BD_{no\ pika} \times T \times (1-\theta_{no\ pika}) \times 0.01 \times 100\% \quad (7)$$
Where $SNSi_{no\ pika}$ was soil total nitrogen, phosphorus, potassium stock in the plot without
plateau pikas (kg m$^{-2}$), $SNi_{no\ pika}$, $BD_{no\ pika}$ and $\theta_{no\ pika}$ were soil nutrient concentration (g kg$^{-1}$),
soil bulk density (g cm$^{-3}$) and soil fraction of gravel larger than 2 mm in plots without plateau
pikas, respectively; and $T$ was soil thickness (20 cm).
**2.6 Data analysis**
Data from 50 disturbed plots and 50 undisturbed plots were used to examine the
difference in ecosystem services of alpine meadows between the presence of plateau pikas
and the absence of plateau pikas; and then data from 50 disturbed plots were used to examine
the responses of each ecosystem service of alpine meadows to the disturbance intensity of
plateau pikas.
All data variables (palatable plant biomass, plant-species richness, soil-water storage,
soil organic carbon stock, soil total nitrogen stock, soil total phosphorus stock, and soil total
potassium stock) were assessed for the normality and homogeneity by using the Shapiro-Wilk
test. If necessary, the data were base-10 log-transformed to fit the assumption of normality
and homogeneity for further variance analysis.

A Linear Mixed Model (LMM) with the function "lmer" from the lme4 package was

used to examine differences in palatable plant biomass, plant-species richness, soil-water
storage, soil organic carbon stock, soil total nitrogen stock, soil total phosphorus stock, and
soil total potassium stock between the presence and absence of plateau pikas across the five
sites. In linear mixed models, the abovementioned parameters acted as response variables, the
absence/presence were introduced as fixed factor, and the paired plots nested within each site
as a random factor.

To clarify the responses of palatable plant biomass, plant-species richness, soil-water

storage, soil organic carbon stock, soil total nitrogen stock, soil total phosphorus stock, and
soil total potassium stock to the disturbance caused by plateau pikas, a linear model (LM) was
used to examine the relationships between these variables and active burrow entrance
densities in all plots with plateau pikas. The densities of active burrow entrances by plateau
pikas were considered to be the fixed factor, and were used to construct the regression
analysis between palatable plant biomass, plant-species richness, soil-water storage, soil
organic carbon stock, soil total nitrogen stock, soil total phosphorus stock, and active burrow
entrances densities. To select the final regression models, likelihood ratio tests were used to
compare simple linear regression and polynomial regression models. After likelihood ratio
tests, the models with $p < 0.05$ and the smaller Akaike Information Criterion (AIC) were used
as the final regression models.

The Bonferroni's test used to adjust *P* values and made to correct for experiment-wise

error rates. All statistical analyses were performed with R 4.0.2 (R Foundation for Statistical
Computing, Vienna, Austria).
**3 Results**
**3.1 Effects of the presence of plateau pikas on the ecosystem services of alpine meadows**

The palatable plant biomass (Fig. 1A) and soil-water storage (Fig. 1B) were 19 % and

16 % lower in the plots with plateau pikas than in the plots without plateau pikas, whereas
soil organic carbon stock (Fig. 1C), plant-species richness (Fig. 1D), soil total nitrogen (Fig.
1E) and total phosphorus stocks (Fig. 1F) in the plots with plateau pikas were 29 %, 15 %,
10 % and 8.9 % higher than those in the plots without plateau pikas. In addition, there was no
difference in the soil total potassium stock between the plots with and without plateau pikas
(Fig. 1G).

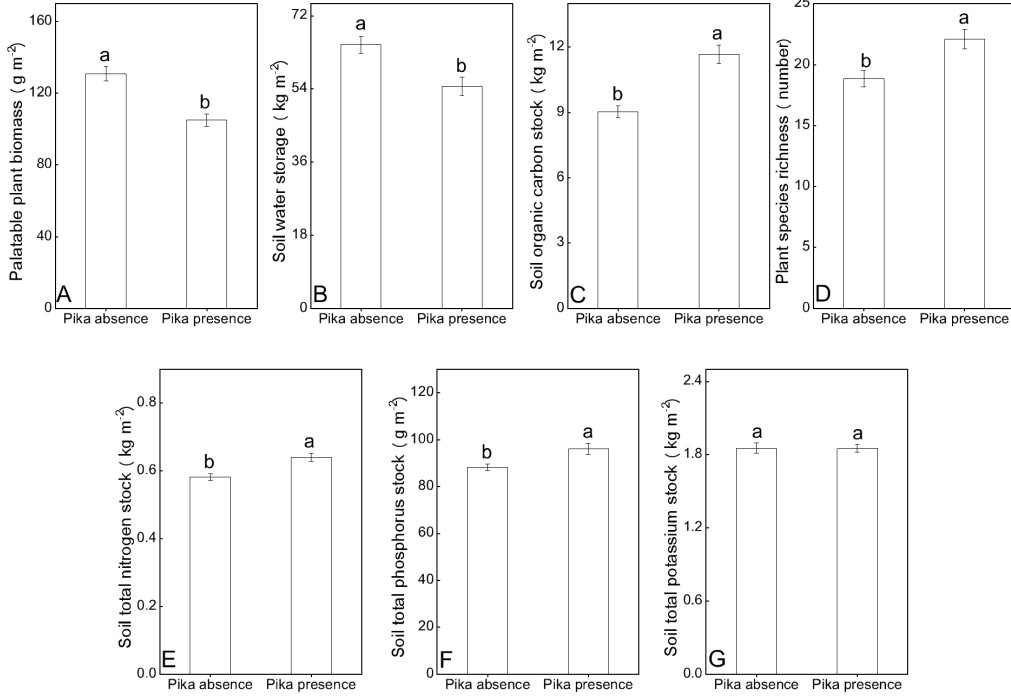


**Figure 1.** Palatable plant biomass (A, F = 46, $p < 0.001$), soil-water storage (B, F = 35, $p <$
0.001), soil organic carbon stock (C, F = 88, $p < 0.001$), plant-species richness (D, F = 64, $p =$
0.003), soil total nitrogen stock (E, F = 22, $p < 0.001$), soil total phosphorus stock (F, F = 12,
$p = 0.004$), and soil total potassium stock (G, F = 0.03, $p = 0.88$) of plots with and without
plateau pikas (mean ± standard error). Lower case represents a significant difference between
the absence and presence of pika based on an LMM.
**3.2 Effects of plateau pikas' disturbance intensity on the ecosystem services of alpine**
**meadows**
The palatable plant biomass (Fig. 2A), soil organic carbon stock (Fig. 2C), plant-species
richness (Fig. 2D), soil total nitrogen (Fig. 2E), and phosphorus (Fig. 2F) stocks had the
maximum values as the plateau pikas' disturbance intensity increased. While the soil-water
storage of the topsoil layer (Fig. 2B) decreased linearly with increasing disturbance intensity
of plateau pikas. In addition, the disturbance intensity of plateau pikas had no obvious
relationship with soil total potassium (Fig. 2G).

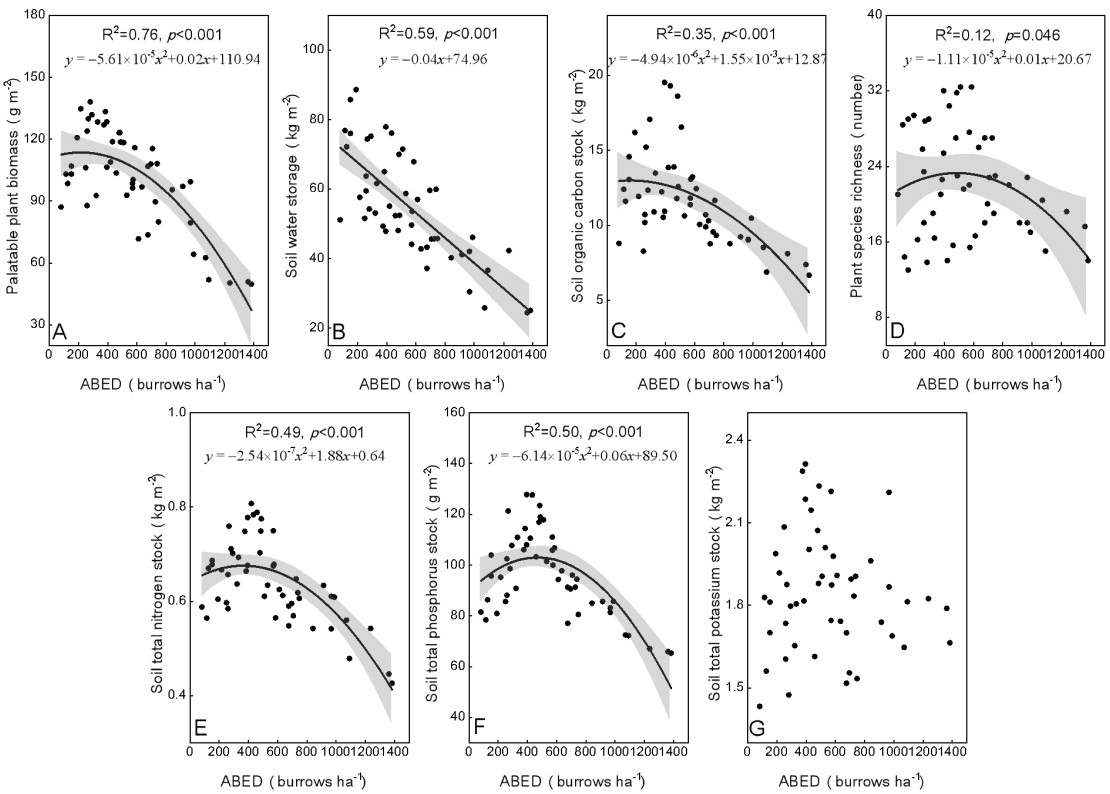


**Figure 2.** The palatable plant biomass (A, F = 69), soil-water storage (B, F = 69), soil organic

carbon stock (C, F = 13), plant-species richness (D, F = 3.3), soil total nitrogen stock (E, F =

23), soil total phosphorus stock (F, F = 24), soil total potassium stock (G) for different

disturbance intensity of plateau pikas based on linear models (LMs). An adjusted local

smoothed regression line (black) with its 95 % confident interval (gray) was used to

determine the relationship between the disturbance intensity of plateau pikas and the above

indicators. ABED: active burrow entrance densities

**4 Discussion**

Prairie dogs and European rabbits have been shown to affect grassland ecosystem

services in arid and semi-arid regions (Delibes-Mateos et al., 2011; Martínez-Estévez et al.,

2013). This study combined the home-range scale and a quadrat scales to test how the

presence of plateau pikas and its disturbance intensity influence the ecosystem services of

alpine meadows, and found that the presence of plateau pikas and its disturbance intensity
indeed impacts the ecosystem services of alpine meadows, similar to prairie dogs and
European rabbits in grassland ecosystem services in arid and semi-arid regions.
Lower palatable plant biomass in the presence of plateau pikas indicates that the
presence of plateau pikas reduces the forage available to livestock, which is consistent with
the results of European rabbits in semi-arid regions (Eldridge and Myers, 2001;
Delibes-Mateos et al., 2008), and is not consistent with results from prairie dogs in arid
regions (Martínez-Estévez et al., 2013). Prairie dogs benefit perennial plants in arid
grasslands, in which blue gramma (*Bouteluoa gracilis*) and vine mesquite (*Panicum obtusum*)
are palatable perennials for livestock (Sierra-Corona et al., 2015), whereas European rabbits
increase unpalatable plants (*Marrubium vulgare* and *Colchicum melitensis*) because they
prefer grasses (Leigh et al., 1989; Eldridge and Myers, 2001). Plateau pikas enable more
unpalatable broad-leaved plants to grow in alpine meadows (Pang and Guo, 2018) and can
bury many plants (Pang and Guo, 2017). However, their consumption patterns can benefit the
growth of palatable plants (Pang and Guo, 2017), because plateau pikas preferentially
consume unpalatable dicotyledons (Zhao et al., 2013; Pang and Guo, 2017). The tradeoff
between the decrease and increase in palatable plant biomass contributes to a negative effect
on palatable plant biomass on a home-range scale, resulting in a decrease in the forage
available to livestock. These results demonstrate that the presence of small mammalian
herbivores affects the forage available to livestock of grassland ecosystems may be related to
environmental conditions. Specific performance is that the presence of small mammalian
herbivores is disadvantageous to the forage available to livestock in semi-arid and alpine
regions, but it is beneficial to forage available to livestock in arid regions.
The presence of plateau pikas has different impacts on regulating services of alpine
meadows, when assessed by different indicators. The presence of plateau pikas leads to lower
soil-water storage, resulting in a decrease in the water conservation, whereas the presence of
plateau pikas can lead to higher soil organic carbon stock, implying that the presence of
plateau pikas can increase the carbon sequestration. Lower water conservation of alpine
meadows in relation to the presence of plateau pikas is consistent with the effect of European
rabbits' presence on the water conservation of grasslands in semi-arid regions (Eldridge et al.,
2010), whereas it is inconsistent with the presence of prairie dogs in relation to the water
conservation in arid regions (Martínez-Estévez et al., 2013). This difference in ascribed to
evaluation indicators for the water conservation. The water infiltration rate is considered as an
index to evaluate the effect of prairie dogs on the water conservation of grasslands in arid
regions (Martínez-Estévez et al., 2013). In contrast, the water storage of topsoil is used to
evaluate the effects of European rabbits and plateau pikas on the water conservation of
grasslands in semi-arid grassland and alpine meadow (Eldridge et al., 2010). The activities of
European rabbits and plateau pikas can reduce the crust cover of grasslands and increase
water infiltration from top soil to deep soil in semi-arid regions (Eldridge et al., 2010; Li et al.,
2015), contributing to a negative effect on the water conservation in the topsoil layer. This
study shows that the presence of plateau pikas leads to higher the carbon sequestration in
alpine meadows, similar to the effect of the presence of prairie dogs in arid regions
(Martínez-Estévez et al., 2013) and European rabbits in semi-arid regions (Delibes-Mateos et
al., 2011). Plateau pikas can input extra organic matter through the deposition of uneaten food
(Liu et al., 2009; Zhang et al., 2016; Yu et al., 2017a) and the excretion of urine and feces
(James et al., 2009; Yu et al., 2017b), which increases the soil organic carbon stock and
contributes to an increase in the carbon sequestration of alpine meadows. These results
indicate that the presence of small mammalian herbivores can increase the carbon
sequestration of grasslands.

Higher plant-species richness in the presence of plateau pikas shows that the presence of

plateau pikas can lead to higher biodiversity conservation, similar to the effect of European
rabbits in semi-arid regions (Delibes-Mateos et al., 2008) and prairie dogs in arid regions
(Davidson et al., 2012). The mechanisms by which small mammalian herbivores lead to
higher plant-species richness have been discussed in many previous studies (Zhang et al.,
2020; Pang et al., 2021b). The presence of plateau pikas can lead to higher soil total nitrogen
and total phosphorus stocks, demonstrating that plateau pikas can increase the soil nitrogen
and phosphorus maintenance. In addition, there was no difference in the soil total potassium
stock between the areas with and without plateau pikas, indicating that the presence of plateau
pikas had no effect on the soil potassium maintenance. This effect was also observed with
prairie dogs and European rabbits in arid (Delibes-Mateos et al., 2011) and semi-arid regions
(Delibes-Mateos et al., 2008; Willott, 2001). Some of the following factors explain the higher
soil nitrogen and phosphorus stocks caused by plateau pikas. The presence of plateau pikas
can increase the input of soil organic material (Liu et al., 2013; Zhang et al., 2016; Pang et al.,
2020a). Secondly, the presence of plateau pikas can result in higher organic nitrogen and
phosphorus stocks (Yu et al., 2017b), which contributes to higher soil nitrogen and
phosphorus maintenance. These results suggest that a general pattern can be identified
regarding the effect of the presence of small mammalian herbivores on the supporting
services of biodiversity conservation, soil nitrogen, and phosphorus maintenance.

This study also shows that the disturbance intensity of plateau pikas also affects the

forage available to livestock, biodiversity conservation, water conservation, carbon
sequestration, and soil total nitrogen and phosphorus maintenance, and these effects is related
to disturbance intensity of plateau pikas. In this case, the active burrow entrances caused by
plateau pkas was used to indicate the all disturbance intensity of plateau. However, active
burrow entrances in disturbed plots was greatly changeable. This study just uses the field
survey data in this experiment to simulate the effect of disturbance intensity of plateau pikas
on the palatable plant biomass, soil organic carbon stock, plant-species richness, soil total
nitrogen, phosphorus and potassium stocks. As found in plant-species richness and
aboveground plant productivity (Dial and Roughgarden, 1998; Gao and Carmel, 2020), the
response of plant-species richness and palatable plant biomass to the disturbance intensity of
plateau pikas follow the pattern for the intermediate disturbance hypothesis in this study. In
addition, the soil organic carbon stock, soil total nitrogen and phosphorus stocks at
home-range scale also support the intermediate disturbance hypothesis. However, the top soil
water storage does not conform the intermediate disturbance hypothesis.

At lower disturbance intensity, stronger competition of dominant sedges often restrains

the grass to grow well (Pang and Guo, 2018) and the rare plants to coexist (Wang et al., 2012),
which leads the forage available to livestock and biodiversity conservation of alpine meadows
to be maintained at a low level. Although the presence of plateau pikas can increase the input
of soil organic matter, this increase is low (Pang and Guo, 2017; Pang et al., 2020b), which
enables the soil organic carbon sequestration and soil nitrogen and phosphorus maintenance
of alpine meadows to maintain a relatively low level.
At intermediate disturbance intensity, the activities of plateau pikas improve the growth
potential of grass plants (Wang et al., 2012), and increase the input of organic matter, soil total
nitrogen (Li et al., 2014), organic carbon accumulation (Yu et al., 2017b), which contributes
to higher the biodiversity conservation, forage available to livestock, carbon sequestration,
soil total nitrogen and phosphorus maintenance services.
At higher disturbance intensity of plateau pikas, frequent bioturbation can enable all
species to be at risk of going extinct (Dial and Roughgarden, 1998). Low soil water content in
alpine meadows (Liu et al., 2013) only sustains the xerophytes and mesophytes, most of
which are unpalatable (Pang and Guo, 2018). This contributes to relatively lower forage
available to livestock and biodiversity conservation. Low vegetation biomass decreases the
input resources of soil organic matter (Sun et al., 2015; Pang and Guo, 2017), contributing to
a decrease in the soil organic carbon sequestration and soil nitrogen and phosphorus
maintenance of alpine meadows.
Additionally, the linearly negative relationship between the water conservation of alpine
meadow and disturbance intensity of plateau pikas is ascribed to evaporation and more water
infiltration on bare soil patches, as the amount of water evaporation and infiltration tends to
increase as the area of bare soil increases (Liu et al., 2013).
Together with previous studies (Delibes-Mateos et al., 2011; Martínez-Estévez et al.,
2013), this study demonstrates that the presence of small mammalian herbivores has similar
impacts on the biodiversity conservation, soil nutrient maintenance, and carbon sequestration

of grasslands throughout the arid, semi-arid, and alpine regions, whereas the effects of the presence of small mammalian herbivores on the forage available to livestock and water conservation are dependent on environmental conditions. This study further verifies that the disturbance intensity of plateau pikas also has a significant impact on the ecosystem services of alpine ecosystems. These results concur with the findings in research fields of small mammalian herbivores in relation to grassland ecosystem services.

**5 Conclusions**

This study focused on plateau pikas to investigate the responses of forage available to livestock, water conservation, carbon sequestration, soil nutrient maintenance, and biodiversity conservation of meadow ecosystems to the presence of a small mammalian herbivore and its disturbance intensity across five sites. This will provide insight into the relationship between small mammalian herbivores and ecosystem services of grasslands. The results of this study showed that the presence of plateau pikas led to higher biodiversity conservation, soil nitrogen and phosphorus maintenance, and carbon sequestration of alpine meadows, whereas it led to lower forage available to livestock and water conservation of alpine meadows. Furthermore, this study found that the effect of plateau pikas disturbance intensity on the forage available to livestock, biodiversity conservation, soil maintenance of nitrogen and phosphorus, and carbon sequestration also conformed to the moderate disturbance hypothesis. These results verified that plateau pikas could affect the ecosystem services of meadow ecosystems in alpine regions and present a relatively complete pattern of small mammalian herbivores influencing grassland ecosystem services.

*Author contributions.* YC and ZG conceived the ideas and designed the methodology; YC, XP,
GB and HY collected the data; YC analysed the data; YC and ZG wrote the manuscript. All of
the authors contributed critically to the drafts and gave their final approval for publication.

*Competing interests.* The authors declare that they have no known competing financial
interests or personal relationships that could have influenced to influence the work reported in
this paper.

*Acknowledgments.* The authors would like to thank Jing Zhang, Qian Wang, Haipeng Xu,
Wenna Zhang, Juan Wang, Ding Yang, Jie Li, Fuyun Qiao, Digang Zhi, Haohao Qi, Ganlin
Feng and Yuanyuan Duan from Lanzhou University for the contributions made to this study
through their field assistance and laboratory analysis.

*Financial support.* This study was funded by the National Natural Science Foundation of
China (32171675), the Key Laboratory of Superior Forage Germplasm in the Qinghai-Tibetan
Plateau (2020-ZJ-Y03) and the Open Project of State Key Laboratory of Plateau Ecology and
Agriculture, Qinghai University.

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
