# Peer review of "Effect of the presence of plateau pikas on the ecosystem services of alpine meadows"

_Biogeosciences, 2021_

## Referee Report (RR1)

Small mammal herbivores often cause extensive disturbance to the grassland and may affect the ecosystem service function of the grassland ecosystem. Taking the plateau pika commonly found in Qinghai-Tibet alpine meadow as an example, the manuscript studied the interference of small mammals' herbivores and the effect of interference intensity on ecosystem service function in Qinghai-Tibet alpine meadow. It was found that different alpine meadow ecosystem service function evaluation indexes were different, and the existence of plateau pika and its interference intensity had different effects on alpine meadow ecosystem service function. These findings could improve the understanding of the relationship between small mammals, herbivores and grassland ecosystem services. The design of the experiment is reasonable and the effect is remarkable. In my opinion, its theme is our journal that its readers are interested in. The author makes full revision in accordance with the comments and suggestions put forward by the review experts and editors, and the manuscript can make a significant contribution to scientific progress within the scope of this journal, and discuss the results in an appropriate and balanced manner. Scientific results and conclusions are clear, concise, well structured. I suggest the manuscript can be published.

---

## Author Response (AR3)

Dear Editor

  Thank you for your patience with our manuscript numbered "BG-2021-313". Based on the further comments from the reviewer and editor, we have revised the manuscript carefully again. We found that these questions and comments are valuable to improving the quality of this manuscript. Here, we submit both a clean and a track changed versions of manuscript to "Biogeosciences".

  The responses to the reviewer' comments are following:

1. Three significant digits are probably more appropriate for values in the abstract and elsewhere; excessive precision is distracting.

**Answer: Thanks for your comment. We have used two significant digits to revise the values in abstract and results.**

2. On line 30 I'm not sure what 'richening the small mammalian herbivores' means. 'enriching our understanding'?

**Answer: Great comment. We revised the "richening the small mammalian herbivores in relation to grassland ecosystem services" into "enriching our understanding of the small mammalian herbivores in relation to grassland ecosystem services".**

3. On line 33, please note a number of relevant references in this week's Science magazine that emphasizes the importance of grassy biomes globally, https://www.science.org/journal /science

**Answer: Thank you for your comment. We have added the relevant references in this week's Science magazine that emphasizes the importance of grassy biomes globally.**

4. 238 and elsewhere: add non-breaking spaces between values and units

**Answer: Thanks for your comment. We have added non-breaking spaces between values and units throughout the manuscript.**

5. Figure 2 and elsewhere: I'm not fully convinced that a parabolic model is appropriate because it's unclear if the mechanistic response is necessarily parabolic. I can see an increase and then decrease in response to the independent variable here, although no indication that it is necessarily symmetric. A brief justification of this choice of model would be forthcoming (noting that it does align with intermediate disturbance concepts.

**Answer: Good comment. The simulation curves were not necessarily symmetric, and this is related to disturbance intensity of plateau pikas. In this field survey experiment, we could not collect all disturbance intensity of plateau (from 0 to maximum of active burrow entrances caused by plateau pikas). In the revision, we added the final regression models. The likelihood ratio test was used to select the final regression modes in Figure 2, including the palatable plant biomass (Fig. 2A), soil organic carbon stock (Fig. 2C), plant-species richness (Fig. 2D), soil total nitrogen (Fig. 2E), and phosphorus (Fig. 2F) stocks. These final regression models showed that disturbance intensities of plateau pika had a clear maximum value for the palatable plant biomass, soil organic carbon stock, plant-species richness, soil total nitrogen and phosphorus stocks, which was align with intermediate disturbance concepts. How to use the likelihood ratio test to select the final regression modes was explained in data analysis sector in original revision. We revised the relevant expression in abstract and results, and added a brief justification in the discussion.**

The manuscript has been revised carefully and strictly according to your comments. We hope our modification and explanation is clear enough, however, if there is still any question, please do not hesitate to contact us.

Yours sincerely
Ying Ying Chen, Huan Yang, Gen Sheng Bao, Xiao Pan Pang, Zheng Gang Guo*
First author: Ying Ying Chen, Email: chenyy2019@lzu.edu.cn
Corresponding authors: Zheng Gang Guo, E-mail: guozhg@lzu.edu.cn

---

## Author Response (AR4)

**Reviewer 1**

Small mammalian herbivores often create extensive disturbance on grasslands, and might affect the ecosystem services of grassland ecosystem. This study uses plateau pika as an example herbivore to investigate the effect of disturbance by small mammalian herbivores and its disturbance intensity on ecosystem services of alpine meadows on the Qinghai-Tibetan Plateau. This study finds that the presence of plateau pikas and its disturbance intensity have different effect on ecosystem services of alpine meadows when the indicators, which are used to estimate the ecosystem services of alpine meadows, are various. These findings can improve the understanding of small mammalian herbivores in relation to grassland ecosystem services. The experimental design is sound and results are striking. In my opinion, its topic is of interest to its audience of our journal. I suggest that the manuscript can be published after some minor corrections.

**Answer: Thank you for your positive comments, and we have revised the manuscript carefully according your advices.**

General comments:

1. The plateau pika had been introduced in introduction sector and Field survey design sector. It had better move the description of plateau pikas in Field survey design sector to introduction.

**Answer: Good comment. We have moved the description of plateau pikas in Field survey design sector to introduction sector.**

2. The confidence intervals are required to represent in the Figure 2.

**Answer: Thank you for your comment. We have added the confidence intervals in the Figure 2.**

Some specific comments:

1. Line 41: Please add the Latin name of prairie dogs when it first appeared.

**Answer: Thanks for your advice. We have added the Latin name of prairie dogs when it first appeared (Line 41).**

Line 44: What is the Latin name of European rabbit.

**Answer: Thank you for your suggestion. We have added the Latin name of prairie dogs when it first appeared (Line 44).**

Line 141: The same alpine meadow is not good words. It may be alpine meadow with same dominant plant.

**Answer: Thanks for your comment. We have revised "the same alpine meadow" into "the alpine meadow with same dominant plant".**

Line 190-194: The quantity of samples in each plot with or without plateau pika is better to expressed consistently with Arabic numerals?

**Answer: Thank you for your advice. We have revised the quantity of samples in each plot with or without plateau pika with Arabic numerals.**

Line 190-194: 5 soil samples in each plot were mixed into composite sample to measure carbon and nutrient concentrations, or each of 5 soil samples was used to measure carbon and nutrient concentrations. Please clarify it.

**Answer: Good question! In this study, we collected one soil sample from one subplot. 5 samples in each plot from 5 subplots were individually measured. The average value of five soil samples in one plot was considered as the representative data of that plot. We had clarified it in manuscript.**

Line 255: Please move "Differences were considered significant at $p < 0.05$." into the analysis of "LMM" in paragraph 2 of Data analysis sector.

**Answer: Thank you for your suggestion. We have moved "Differences were considered significant at p < 0.05." into the analysis of "LMM" in paragraph 2 of Data analysis sector (Line 249).**

Line 302: a quadrat scales?

**Answer: Thanks for your suggestion. We have revised "a quadrat scales" into "the quadrat scales".**

**Reviewer 2**

I suggest that the manuscript can be published after a little correction.

**Answer: We thanked you for your positive comments, and we have revised the manuscript carefully according your advices.**

1. "this study" appeared in the abstract four times. It is necessary to further polish the language.

**Answer: Thank you for your advice. We have polished the language in the abstract.**

2. For references in scientific hypothesis, please check whether there is the necessity of citation. So, it is suggested to summarize the existing research in the research progress. Then put forward scientific hypothesis.

**Answer: Exactly, we had summarized the existing research progress in front paragraphs of introduction. It is unnecessary to cite the references in hypothesis, and it seems redundant. To make the hypothesis more scientific, we have deleted the references in the hypothesis according to your opinion.**

3. P96-97 "Plateau pikas can live in various habitats with different soil types, topographies, and microclimates." This sentence lacks qualified region.

**Answer: Thanks for your suggestion. We have added the qualified region "on the Qinghai-Tibetan Plateau" in this sentence (Line 97).**

4. What is the grazing situation of the experimental site and how to eliminate the impact of grazing intensity on the grassland.

**Answer: Thanks for your comment. This study used a stratified random and paired design to discover the general effect of plateau pika disturbance on ecosystem services of alpine meadows. In experimental design, each paired plot shared the same grazing intensity during the cold season, however, 50 paired plots consisted of different yak grazing intensity, and this can permit the general pattern relating to the effect of plateau pika disturbance on alpine meadow ecosystem services. We have supplemented this information into "Field survey design" sector.**

5. ABED (active burrow entrance densities), when is the investigation period?

**Answer: Thanks for your comment. The ABED (active burrow entrance densities) was measured in the process of field sampling. As described in field sampling, this study firstly measured the density of active burrow entrance, and then measured the area of bare soil patches, collected vegetation and soil samples.**

6. In Fig.1, the presence of plateau pika, What pika density are based on?

**Answer: Thanks for your comment. Fig.1 shows that plateau pika density is a qualitative description, which is to present the difference in ecosystem services of alpine meadows between the presence of plateau pikas and the absence of plateau pikas. Fig.2 shows that**

**the changeable trends of each ecosystem service of alpine meadows as the disturbance intensity of plateau pikas increased. In Fig.1, 50 disturbed plots with different plateau pika densities (it ranged from 83 to 1384 entrances ha$^{-1}$) was considered as a whole to compare with the 50 undisturbed plots. We supplement some sentences to clarify this research approach in data analysis sector.**

**Reviewer 3**

I am in receipt of an additional review from an expert Referee, pasted below. When preparing a response, please include a response to these insightful comments, which follow.

Overall, I am very excited to see the authors addressing this topic in this system. I especially appreciated the authors integrating a number of measures into their suite of response variables; that should be done more frequently! On the other hand, some of the methods either need to be clarified or better justified, because the current description comes across as potentially biased. Furthermore, I don't understand why the authors make things (for me, at least) more complex and difficult to understand – instead of simply naming the particular response variable, they have created monikers such as "biodiversity conservation service" that obfuscate meaning and make reading harder. Also, as noted below, I'm not sure that I agree with the authors' assertion of a threshold, simply because a quadratic relationship exists. Isn't that the default assumption of niche dynamics and (more relatedly) the Intermediate Disturbance Hypothesis? I don't see where the bare vs. the vegetated plots in the pika-occupied areas is discussed in the main text. Also, some of the Discussion seems like a pretty strong over-simplification of how to contextualize the study's results with the broader literature. Also, given that all of the sampling occurred in 1 month of 1 year, and that the entire sampled area spanned 0.1225 km2, it would seem appropriate for the authors to acknowledge with a caveat or two the limitations of this spatio-temporal domain. I realize that the sampling scales of biogeochemistry investigations are much more limited than that of animal ecologists, but note that you're investigating the dynamics of both soils and animal behaviors.

**Answer: We are appreciated for your comments which are really helpful for improving our manuscript. There are many comments in this paragraph. We will respond these comments as following one by one.**

Comment 1: some of the methods either need to be clarified or better justified, because the current description comes across as potentially biased. **This comment is not specific; therefore, we cannot do anything here. In the next comments, we can clarify and justify the methods when any comments relate to methods.**

Comment 2: I don't understand why the authors make things (for me, at least) more complex and difficult to understand – instead of simply naming the particular response variable, they have created monikers such as "biodiversity conservation service" that obfuscate meaning and make reading harder. **We are sorry for moniker name of biodiversity conservation service and its name is ecological service of biodiversity conservation. When we had invited English Language Editing Services to polish this**

manuscript, the editor kept it. In the revision, we have revised "biodiversity conservation service" into "ecological service of biodiversity conservation".

Comment 3: Also, as noted below, I'm not sure that I agree with the authors' assertion of a threshold, simply because a quadratic relationship exists. Isn't that the default assumption of niche dynamics and (more relatedly) the Intermediate Disturbance Hypothesis? **Great comment! It is very accurate that the threshold is a not good expression. In china, plateau pika is considered as a pest and government costs big money to eradicate it every year. We had intended to provide some advices for government to control plateau pikas, rather than eradicate it. Therefore, we propose a threshold for plateau pikas, in which plateau pikas can play positive roles in alpine meadows. Now, we think about it carefully, we just found the Intermediate Disturbance Hypothesis in ecology, and we cannot present a threshold for plateau piaks. Therefore, in revision, we have deleted the threshold description throughout the whole manuscript.**

Comment 4: I don't see where the bare vs. the vegetated plots in the pika-occupied areas is discussed in the main text. **Previous studies have shown that soil nutrient concentration and stock are different between bare soil patches and vegetated surface areas (Pang et al., Geoderma, 2020, 372, 114392; Yu et al., Geoderma, 2017, 307, 98-106). Therefore, this study used plot scale (home range) to whether the presence of plateau pika has an impact on grassland ecosystem services, in which, the percentage of soil bare area and vegetated area in each plot with plateau pikas was to comprehensively calculate the soil nutrient stock and plant biomass at plot level. The difference in nutrient and plant biomass between bare soil patches and vegetated surface areas is not key points, which has been explained in Method sector. This is to say that this study focuses on the plateau pikas disturbance, rather than bioturbation.**

Comment 5: Also, some of the Discussion seems like a pretty strong over-simplification of how to contextualize the study's results with the broader literature. **This is a very common question, and it is suitable for any manuscript. We tried to look up literatures for deepen the discussion. Do hope that our revision is acceptable for associate editor and this reviewer.**

Comment 6: also, given that all of the sampling occurred in 1 month of 1 year, and that the entire sampled area spanned 0.1225 km2, it would seem appropriate for the authors to acknowledge with a caveat or two the limitations of this spatio-temporal domain. **Plateau pikas is social animal and adapts its habitats. We selected five sites to test the same question and hope to find a general pattern. We are sure that this is similar to other studies (Li et al., Frontiers in plant science, 2021: 2455; Wang et al., Frontiers in plant science, 2022, 13: 830856). This manuscript is to find a general effect of plateau pika disturbance on ecosystem services of alpine meadows, bot focuses on spatio-temporal domain.**

Comment 7: I realize that the sampling scales of biogeochemistry investigations are much more limited than that of animal ecologists. but note that you're investigating the dynamics of both soils and animal behaviors. **Thank you for your comment. The sampling scales in this study has been widely applied into other study cases (Pang et al., Geoderma, 2020, 372, 114392; Pang et al., European Journal of Soil Science, 2020, 71, 706-715). This sampling scales is enough to test the plateau pika disturbance in relate to**

**soil and plant. If we focus on the plateau pika behaviors, the plot size has to become bigger, such as 50 m × 50 m or 100 m × 100 m. Because this study is to analyze the plateau pika disturbance, rather than plateau pika behaviors, we think that the sampling scale is acceptable.**

Larger, more-important topics that are of greatest concern to the robustness of the study and its conclusions:

1. Line 27 – If this is true, consider adding "linearly", immediately after "decrease". Later on in the sentence, note that nowhere in the abstract have you defined what "disturbance" or "disturbance intensity" is, specifically. This 2nd point remains true on line 54 through at least 66, in the Introduction. It appears again on line 152. What is the "disturbance"? Needs to be defined.

**Answer: Thanks for your advice. We have added the "linearly" after "decrease" in Line 27. "disturbance" is to be defined to the all activities of plateau pika, including the developing burrow systems, excreting feces and urine, consuming plants, clipping tall plants, and producing bare soil patches. We have added this information into manuscript. "disturbance intensity" is to be defined to the plateau pika populations, which is measured by the density of active burrow entrances. We have presented this information in Method (Line 175). The disturbance intensity of plateau pikas is a common word in this field. Here, we used the disturbance intensity is to simplify the disturbance intensity of plateau pikas, which has been edited by Language services. Now we have revised "disturbance intensity" into "the disturbance intensity of plateau pikas" throughout the manuscript.**

2. Line 32-33 and 70-71 – It's not clear what "provisioning, regulating, supporting" refer to: each needs to be in reference to something else. E.g., provisioning WHAT?, regulating WHAT?, supporting WHAT?

**Answer: Thank you for your comment. In the process of revision, we have added the specific details about provisioning, regulating, supporting and cultural services in introduction.**

3. Line 49-51 – This sentence needs to be re-phrased; its meaning is not clear, and there are several studies (in just 1 species of pika [Ochotona princeps], alone) that show how American pikas are ecosystem engineers, alterers of vegetation composition, a keystone species, etc. These include Aho et al. (1998), several papers by Denise Dearing at the University of Utah (see her thesis: https://www.proquest.com/docview/304226940?pq-origsite=gscholar&fromo penview=true), Dearing (1996: Oecologia), Hall and Chalfoun (2019; J. Animal Ecology), Jakopak et al. (2017, J. Mammalogy), among others. Also, it would be important for the authors to cite one or more papers that have noted specifically that the plateau pika is a keystone species. This will help ensure the objectivity and lack of bias in the research, rather than it appearing that plateau pikas are nothing more than a "pest" (see N. Fan, W. Zhou, W. Wei, Q. Wang, and Y. Jiang. 1999. chapter 13. Rodent Pest Management in the Qinghai-Tibet Alpine Meadow Ecosystem. 20 pages.). This lack of clarity here is pivotal, because this is where you're really setting up the goal of the manuscript and why it will be an important contribution.

**Answer: As the reviewer says, several studies show how American pikas (*Ochotona***

*princeps*) **are ecosystem engineers, alters of vegetation composition, as the keystone species. However, American pika are the rock-dwelling, whereas plateau pika are meadow-dwelling (Smith et al., Lagomorph biology, 2008: 89-102.). in addition, no study is to test the effect of American pikas on grassland ecosystem services. At present, the European rabbit and prairie dog was verified to have impact on grassland ecosystem services in arid and semiarid regions (we have presented this information in introduction sector). Delibes-Mateos et al (Delibes-Mateos et al., Biological Conservation, 2011, 144(5): 1335-1346.) review the small rodent in relate to soil and vegetation, just speculate that small rodents possibly affect the ecosystem services, but this paper does not quantify the effect of small rodent on ecosystem services of grasslands. This study used five site data to test the effect of plateau pika (*Ochotona curzoniae*) on meadow ecosystem services in alpine regions, which will richen the small mammalian herbivores in relation to grassland ecosystem services.**

**There are many papers to describe the role of plateau pika in alpine meadows (Smith and Foggin, Animal Conservation, 1999, 2(4): 235-240; Lai and Smith, Biodiversity & Conservation, 2003, 12(9): 1901-1912; Delibes-Mateos et al., Biological Conservation, 2011, 144(5): 1335-1346; Sun et al., Grassland Science, 2015, 61(4): 195-203; Smith et al., Integrative Zoology, 2019, 14(1): 87-103.), in which plateau pikas are considered as ecological engineering and a keystone species, or plateau pikas are considered as a pest. We supplemented this information into introduction sector.**

4. Lines 85-93 – I very much like that you have laid out your hypotheses and cited one or more studies that found a certain result. However, unless the reader goes and reads all five of those papers, it's not clear whatsoever why those predictions are being made, nor by which processes or mechanisms those results were created. It would be helpful if you provided concise descriptions of those.

**Answer: Another reviewer suggest that this manuscript had better to delete the references in hypothesis, because it seems redundant. In this revision, we combined two reviewers' comments as following. First, we had summarized the existing research progress with references, and then we present the hypothesis.**

5. Lines 103-105 – Excellent that you provide the reader some overview of conditions at the site. However, it would be much, much more informative, and relevant to your study objectives and interpretation of your results, if you were to provide understanding of how much of the annual precipitation falls as snow (either % or amount during the warm season, and % or amount as snow during cold season), and what the temperatures are in the warm and cold seasons. Pikas generally do not respond to any annual-average measure.

**Answer: Thank you for your comment. We have added the percentage of annual precipitation during the warm season. We presented the average temperatures in the warm season, cold season, and all year.**

6. Lines 114-115 – I'm not sure what it means that "… many plant species are found until late summer."

**Answer: The growing season for plants is short on the Qinghai-Tibetan Plateau, and temperature is low. Different plants are regreened at different time. Some plants are turned green in July due to higher temperate in study regions. This sentence is to say that field survey and sampling in August is optimal because August is good time to**

**identify all plants. We have revised it.**

7. Line 116 – It's not a "census"; it's a "sample" or a "survey".

**Answer: Thank you for your comment. We have revised "census" into "survey".**

8. Line 116-118 – Yak grazing appears that it could be a confounding influence, here.   Given that effects in low-productivity systems such as this are likely to have ecological memory (or legacy effects), this sentence does not make sense logically to me.

**Answer: Thank you for your comment. We have deleted this sentence.**

9. 119 – "only a small burrowing herbivore" … what does this mean? It's not important?

**Answer: There are many small burrowing herbivores on the Qinghai-Tibetan Plateau. However, in this study, the small burrowing herbivores is only plateau pikas in field survey areas. In order not to cause misunderstanding, we think that it necessary to stress it in manuscript. We have revised it as "the small burrowing herbivore at each survey site was only plateau pikas."**

10. Section 2.2 – You might give just a little more background on the life-history strategy of plateau pikas here, as most readers of the journal will not know the relevant details.   E.g., typical body mass, does sexual dimorphism exist?, are they generalist herbivores or if not, what do they eat?, are they burrowers themselves, or do they conscript burrows made by other species?, how deep do their burrows go?, do they hibernate?, etc. May only need to be an extra 2-4 sentences, but this will help the reader immensely.

**Answer: Thank you for your advice. Anther reviewer proposed that introduction of plateau pikas had better present in introduction. We combined two reviewers' comments, and supplemented this information (typical body mass, sex dimorphism, plants that plateau pikas like to eat, how to plateau pika burrows, whether plateau pika hibernate) proposed by this reviewer into introduction.**

11. 123 – When you say "diffusion", do you mean "dispersal"? If so, is this natal dispersal, or adult dispersal? If the process is gradual, does the ability to find reference (unused) sites depend on the of your sampling?

**Answer: Maybe dispersal is correct. We are not native English speaker. The diffusion and dispersal are same to us. Based on reviewer's advice, we changed "diffusion" into "dispersal". Yes, the dispersal is adult. Their babies stay with home together in bearing year. We have revised it. The adult dispersal is not dependent on time, but it is dependent on food resources. In a home range of plateau pikas, when food is insufficient for plateau pika family, and adult plateau pikas often move to a potential habitat with open and low meadows.**

12. 133-134 – I don't understand this sentence at all; the text after the comma is exactly the same as the start of the sentence.

**Answer: Great, it is redundant expression. We have revised it in the revision.**

13. 134-135 – To make the study repeatable, we need to know what that distance was specifically that you used.

**Answer: Good, the distance between the two plots with plateau pikas was more than 3 km, which ensures that the plateau pika of the same family will not appear in two plots with plateau pikas at the same time, the first area with plateau pika presence is found, and it was selected as the first survey plot. And then we continue to find the second area along one direction after 3 km, if alpine meadows with plateau pikas was to design the**

**second plot.**

14. 152 – Depending on what "disturbance" means, this assumption may or may not hold true. I would be surprised if it did NOT hold for amount of biomass within a certain distance of the burrow entrance (if the species is a central-place forager), but as noted earlier in the MS, some ecosystem properties are not affected while others are even promoted by the presence of pikas.

**Answer: Frankly, we did not catch what reviewer say. The plateau pikas lives in their home range with many behaviors. Different activities often occur at different spatial position. Consuming plant was randomly found in whole home range, while clipping plants were only found at edge of the active entrance, however, excreting feces and urine was found at the edge of active and abandoned entrance. The entrance was randomly founded in home range. Here, "disturbance" is to be defined to the all activities of plateau pika. That is to say, disturbance is comprehensive outcome of plateau pika activities, including forager. This is widely reported in previous studies (Pang et al., Geoderma, 2020, 372, 114392; Pang et al., Land Degradation & Development, 2021, 32(3): 1205-1212; Li et al., Frontiers in plant sciewaynce, 2021, 2455; Wang et al., Frontiers in plant science, 2022, 13: 830856). In addition, the density of active burrow entrances was used as a proxy for the disturbance intensity of the plateau pikas has also been widely used in the previous studies (Guo et al., Acta Ecologica Sinica, 2012, 32, 104-110; Liu et al., Plant and Soil, 2013, 366(1): 491-504; Sun et al., Grassland Science, 2015, 61(4): 195-203; Liu et al., Ecological Engineering, 2017, 102: 509-518; Yu et al., Geoderma, 2017, 307, 98-106; Wang et al., Ecological Engineering, 2018, 113: 35-42).**

15. 156-160 – The reader does not have enough detail to know what you are doing, to assign causality by plateau pikas. Rather than assume that the reader will just trust your method, you need to provide clear information that both makes the method repeatable, and convinces the reader that you accounted for this in a robust, defensible manner.

**Answer: Sorry, we did not get this comment core. The alpine meadows with plateau pikas consisted of many bare patches and vegetated surface. We are sure that the soil and plant was different between bare patches and vegetated surface. So, we calculated the nutrient and carbon stock, and plant biomass at plot level, which is correct to compare the areas with and without plateau pikas.**

16. 163-164 – It seems that if you are only moving the quadrats slightly in the pika-occupied sites (but not in the non-pika-occupied sites), you are biasing the sampling and results. Either clarify or justify this approach. By not having any sites be randomly selected, it causes concern in the reader and in my mind as well. Furthermore, in reading the rest of the main body of the text, I'm not seeing any reference to the comparisons of bare vs. vegetated plots within the plateau pika area

**Answer: In this study, the plots with and without plateau pikas were randomly selected; however, the quadrats in each plot was not randomly selected, was selected by "W". The data of five quadrats in each plot was pooled to representative of that plot. There are many kinds of bare patches in alpine meadows. This study focused on bare patches caused by plateau pikas. In plot with plateau pikas, if quadrat was justly covered with the bare patches caused by plateau pikas, we slightly moved the quadrat. if quadrat was justly covered with the bare patches caused by other factors, we did not move the**

quadrat. The comparisons between bare vs. vegetated was not core of this study, which has been reported in other studies (Pang et al., Geoderma, 2020, 372, 114392; Yu et al., Geoderma, 2017, 307, 98-106). Generally, bioturbation was verified by comparing the bare vs. vegetated.

17. 164-165 – What is the purpose of the paired bare patch vs. the vegetated patch? The reader has no idea about why you are doing this?

**Answer: This comment is similar to latest comment. Previous studies have shown that in the presence of plateau pika, there are differences in soil nutrients between bare soil patches and vegetated surface areas (Pang et al., Geoderma, 2020, 372, 114392; Yu et al., Geoderma, 2017, 307, 98-106). Alpine meadows in home range included bare and vegetated surface. We needed to design the quadrat at bare patches and vegetated surface to measure the soil nutrient and carbon concentrations, and these data was used to calculate the soil nutrient and carbon at plot level. In addition, no plants or sparse plants were found on bare patches, we measured plant biomass at bare and vegetated surface, respectively. This design is beneficial to compare the parameters between plots with and without plateau pikas, and reflect the effect of plateau pika disturbance on ecosystem services in alpine meadows. In revision, we have added relative information about the purpose of the paired bare patch vs. the vegetated patch.**

18. 172 – Species richness is simply a tally of the species present in a given area; you mention "their numbers", which leads me to think that you also measured abundance, so that you could quantify measures like evenness and Hill series.

**Answer: Thank you for your advice. Here, species richness is number of plant species. The "their number" is not an accurate word. We changed "their number" into "number of plant species".**

19. 173 – "palatable" and "unpalatable" to which species? Obviously, palatability depends upon the herbivore that one is considering.

**Answer: "palatable" and "unpalatable" are for livestock (It mainly refers to yaks and sheep that often graze on alpine meadows) in this study. We have refined this expression in revision.**

20. 192 – You need to connect this back to your sampling approach. Consider adding "Because pika-absent sites did not include sampling at bare areas, only" before the start of the "Five soil samples…" sentence.

**Answer: Great! We completely agreed with your comments. We have added the expression in revision.**

21. 200-201 – What does "artificially removed" mean? I think that you may mean "manually picked out"…?

**Answer: Yes, you are right. In the revision manuscript, we have revised "artificially removed" into "manually picked out".**

22. 202-203 – This sentence is nonsensical, as written – passing soil through a sieve does not allow one to estimate any of these concentrations.

**Answer: Sorry, this is a bad expression. We changed "Finally, the soil organic carbon, nitrogen, phosphorus, and potassium concentrations were determined by passing through a 0.15 mm sieve" into "Finally, soil samples were sieved at 0.15 mm to analyze soil organic carbon, nitrogen, phosphorus, and potassium concentrations in the**

**laboratory.”**

23. 203-206 – Please specify which technique is associated with which concentration – e.g., Kjeldahl procedure measures total N concentration.

**Answer: Thank you for your comment. We have revised the specify technique into “Soil organic carbon was measured using the dichromate heating-oxidation. Soil total nitrogen concentration was measured using the Kjeldahl procedure. Soil total phosphorus concentration was measured using the Molybdenum blue colorimetric method. Soil total potassium concentration was measured using flame photometry”.**

24. 216-218 – Yes, but how do you measure the area of a shape that is irregular? Need more details on how you measured areal extent.

**Answer: Thank you for your suggestion. We cited a reference in Chinese to describe the segmentation method. To positively respond to this comment, we have supplemented some specific measurement details in section 2.3 Field sampling. The measurement method is as follows: Each bare soil patch was identified as regular shape or irregular shape. If one bare soil patch was identified as regular shape, such as rectangle, circle, trapezoid, etc; a ruler was used to measure its length, width, height, diameter, upper and lower bottom, and these data was used to calculate the area of that bare soil patch. If one bare soil patch was identified as irregular shape, this bare soil patch was divided into several regular shapes; the areas of these regular shapes were calculated, respectively; the area sum of these regular shapes form irregular bare soil patch was considered as the area of that irregular bare soil patch.**

25. 219-220 – How close to reality was this consideration? Depending on how far from actual truth it is, this assertion worries me.

**Answer: Bare soil patches caused by other factors (no plateau pikas) is simultaneously existed on the vegetated surface in the presence/absence of plateau pikas. To actual quantify the effect of plateau pikas on ecosystem services of alpine meadows, this study only measured the area of bare soil patches caused by plateau pikas, although there exist multiple types of bare soil patches in alpine meadows. Therefore, in each plot without plateau pikas, bare soil areas caused by plateau pikas were considered to be zero.**

26. 225-229 – You cannot assume that your reader is going to read all 3 of these papers. You need to provide the key details, here, to convince the reader that you're doing this robustly.

**Answer: Thank you for your suggestion. We have provided readers with more details by adding the computational formulas in the revision.**

27. 241 – Is the presence vs. absence of plateau pikas considered your “fixed effect” ?

**Answer: What the absence/presence was considered was different in previous studies. Some studies are expressed as predictors (Yu et al., Geoderma, 2017, 307, 98-106; Yang et al., Catena, 2021, 207: 105625), while others are expressed as fixed factors (Pang et al., Geoderma, 2020, 372, 114392; Wang et al., Frontiers in plant science, 2022, 13: 830856). In this study, we originally considered the absence/presence as predictors. In revision, we have revised “predictors” into “fixed factor” according to this comment.**

28. 243 – This is a VERY long sentence. Consider ending the first sentence in the middle of line 247. Also, the phrase from middle of line 247 to 251 is not written correctly – you're not performing a regression analysis between (nor among) all of the response variables.

**Answer: Thank you for your comment. We have ended the first sentence in the middle**

**of line 247. We used "The densities of active burrow entrances by plateau pikas were considered to be the fixed factor, and were used to construct the regression analysis between palatable plant biomass, plant-species richness, soil water storage, soil organic carbon stock, soil total nitrogen stock, soil total phosphorus stock, and active burrow entrances densities." to substitute for originally "the densities of active burrow entrances were considered a fixed factor, and were used to construct regression analysis between palatable plant biomass, plant species richness, soil water storage, soil organic carbon stock, soil total nitrogen stock, soil total phosphorus stock, soil total potassium stock, and active burrow entrance densities." Because we did not perform a regression analysis between all of the response variables, just performed a regression analysis between some response variables which were significantly related to plateau pika disturbance.**

29. 255 – Given the tens to hundreds of different analyses that you're performing across this study, please provide a justification of why you are not correcting for experiment-wise error rates (e.g., Bonferroni stepwise correction). That is, if you perform 100 tests, you will likely have 5 tests that will be "statistically significant", even when there is no pattern nor biological effect whatsoever, just by chance alone.

**Answer: Thank you for your comment. We have added data analysis are as follows: The Bonferroni's test used to adjust $P$ values and made to correct for experiment-wise error rates.**

30. 259-269 – Consider simply reporting the results of what you found, as opposed to giving everything another name for each predictor. However, what you've done makes things more complicated, in my view, because you've lumped several response variables into classes of responses (such as "provisioning services").

**Answer: This is great comment. We further explained what variables predict. They are not experimental results. In revision, we deleted some sentence, which simply the results of what this study found. In addition, we move these sentence into discussion sector for clarifying what variables predict.**

31. 280-283 – I think that it would be preferable to comment on the fit of the linear vs. the quadratic relationship to the data. Also, does a bell-shaped curve unequivocally indicate a "clear threshold for disturbance"? I'm not sure that it does … what about the Intermediate Disturbance Hypothesis as an alternative explanation for the pattern?

**Answer: Thank you for your comment. We have revised this description and supplemented the Intermediate Disturbance Hypothesis as an alternative explanation for the pattern in discussion.**

32. 322 – Although I appreciate the desire to connect plateau pika activity to positive provision of ecosystem services, using terms like "in relation to the forage availability service of grassland ecosystems" makes it confusing for the reader to understand what is really going on in the text. Also, the logic of lines 321-325 is not at all clear to me … what are you saying is the mechanism causing this context-dependence?, i.e., why are the responses different in the two classes of regions?

**Answer: Thank you for your comment. We have revised "in relation to the forage availability service of grassland ecosystems" into "affects the ecological service of forage available to livestock of grassland ecosystems".**

In addition, "the presence of small mammalian herbivores is disadvantageous to the ecological service of forage available to livestock in semi-arid and alpine regions, but it is beneficial to the ecological service of forage available to livestock in arid regions" is the specific performance of "These results demonstrate that the presence of small mammalian herbivores affects the ecological service of forage available to livestock of grassland ecosystems may be related to environmental conditions". We have revised this expression. Besides, the responses different in the two classes of regions have been discussed deeply in line 418-424.

33. 326-335 – I am very impressed that you are trying to contextualize your results amidst some studies from the existing literature, but you are really undercutting the value of these comparisons by virtue of how high-level, superficial, or simplistic that they are. E.g., why are the results consistent with the one study, but not the other? What are the magnitudes of the effects?

Answer: Thank you for your comment. We have further improved the discussion as follows: The difference of water conservation services is related to the selection of evaluation indicators. Martínez-Estévez et al (2013) used the water infiltration rates as an index to evaluate the effect of prairie dogs on water conservation services. However, the effects of European rabbit and plateau pika on water conservation services of semi-arid grassland and alpine meadow by evaluating the water storage of topsoil.

34. 349-350 – This sentence makes no sense, given that you define "biodiversity conservation [service]" as species richness of plants.

Answer: We do not define species diversity protection services as plant species richness, just use plant species richness to evaluate the effect of plateau pika disturbance on biodiversity conservation services (Wen et al., (2013), Plos One, 8, e58432). The higher species richness of plants in the presence of plateau pikas reflect that the presence of plateau pikas is beneficial to biodiversity conservation, which is one of our goals.

35. 364-365 – I am not demanding that you do that for this study, but quite a lot of your interpretation (e.g., see lines 394-397) rests on the assumption that number of active burrows serves as an accurate index for disturbance intensity. It sure would be nice (and, for me, more empirically compelling) to correlate number of active burrows with number of pika-hours spent foraging aboveground, on one or a small number of days throughout the season. The latter would be a much more direct measurement of one process (i.e., forage consumption) that can lead to some of the changes that you are suggesting are imposed by pikas.

Answer: The number of active burrows serving as an accurate index for disturbance intensity of plateau pikas is widely in previous studies (Guo et al., Acta Ecologica Sinica, 2012, 32, 104-110; Liu et al., Plant and Soil, 2013, 366(1): 491-504; Sun et al., Grassland Science, 2015, 61(4): 195-203; Liu et al., Ecological Engineering, 2017, 102: 509-518; Yu et al., Geoderma, 2017, 307, 98-106; Wang et al., Ecological Engineering, 2018, 113: 35-42). We guess that reviewer hope us to use consumption plants by plateau pikas to explain the relationship between disturbance intensity and ecosystem services. It is difficult for us to do it. Plants are affected by many factors, just depend on consumption by plateau pikas. As far as consumption is concerned, plateau pika can reduce the plant biomass, whereas it consumption can stimulate the compensatory growth of plants. We also guess that reviewer focus on consumption plant by plateau pikas. The plateau pikas

**consume plants in morning and sunset. The disturbance intensity is comprehensive results of all activities of plateau pikas. This study is a survey experiment, rather than control experiment.**

36. Lines 389-394 – As mentioned above, this sentence feels like a pretty strong over-simplification of the dynamics of herbivory, given how few studies the authors are citing across the paper, compared to the plethora of studies on the topics listed in this sentence that exist. None of the numerous review articles published over several decades is cited, and the authors are reporting nothing more than directionality.

**Answer: These sentences are the summary of our whole discussion. The relative ideas and academic views were appearance in above-mentioned text. So we did not cite any reference.**

37. Lines 407-409 – The existence of a quadratic relationship to disturbance intensity does not necessarily indicate an existence of a threshold. Much previous ecological literature has been produced on the topic of thresholds; consider consulting it.

**Answer: Thank you for your comment. We have revised the sentence as follow: Furthermore, this study found that the effect of plateau pikas disturbance intensity on ecological services of forage available to livestock, biodiversity conservation, soil maintenance of nitrogen and phosphorus, and carbon sequestration also conformed to the moderate disturbance hypothesis.**

Issues that compromise the clarity, readability, and breadth of audience of the article:

1. Line 15 – "pika" is singular; either say "pikas" or "the plateau pika" (also on line 52, 53, 62, 151, etc.). Also, "an example" of what? Maybe instead say "a model organism" or "a focal organism".

**Answer: Thanks for your suggestion. We have revised "plateau pika" into "plateau pikas or "the plateau pika" throughout the manuscript, revised "an example" into "a focal organism".**

2. Line 17 – "forage availability": does this mean forage available to pikas, livestock, or other herbivores?

**Answer: Thanks for your comment. The "forage availability" means forage available to livestock in this study. We have revised it throughout the manuscript.**

3. Throughout Abstract and entire MS – Using three or more nouns in a row is called "freight-train wording". Its usage makes it very difficult for the reader to divine which noun(s) are acting as adjectives, and which one(s) are acting as a noun. For greater clarity, you need to either 1) hyphenate the nouns that are acting as adjectives, or, preferably, 2) use prepositions to clarify the relationships among the nouns. For example, at this point in the MS, I have no idea what "soil nutrient maintenance services", "soil potassium maintenance service", or "forage availability service" (line 44) is. I recommend the authors implement these clarifying changes, throughout the MS.

**Answer: Thanks for your advice. We have hyphenated the nouns that are acting as adjectives in a row throughout the manuscript.**

4. Line 23 – Change ", whereas it …" to ". In contrast, it …"

**Answer: Thanks for your suggestion. We have revised ", whereas it…" into ". In contrast, it …".**

5. Line 30 – "richen" should be "richening", to be parallel with "influencing"; I will stop

identifying this type of grammatical error here. The MS will be markedly improved and clearer, when all such issues are resolved.

**Answer: Thanks for your comment. We have revised this type of grammatical error throughout the manuscript.**

6. Line 80 – "land-use" should be "patterns of habitat use" or simply "habitat use"; the former refers most commonly to how humans use landscapes for anthropogenic activities. Not sure what "the scales" means, on line 81.

**Answer: Thanks for your comment. We have revised "land-use" into "habitat use". "the scales" means "spatial scale", we have revised this expression.**

**Reviewer 4**

Chen and others explore the role of pika presence at different densities to ecosystem services. The manuscript makes a few interesting points but could be strengthened by focusing more on fundamental ecological theory.

1. I wasn't entirely sure how it was identified that pikas caused some bare patches and not others, for example if the bare patches were caused by past pika activity. Therefore it was unclear to me how controls were used to test the objectives.

**Answer: this is a common comment we met. The soil bare patches caused by plateau pikas is easily to identify because one soil bare patch caused by pika is paired with a visible burrow entrance (Pang et al. Geoderma, 2021, 115098). Other soil bare patches are not paired with visible burrow entrance.**

**The soil bare patches caused by past plateau pika activities can be divided into two types. Some can be gradually restored, and others are still be bare.**

**In this field survey, we firstly identified the soil bare patches caused by plateau pikas though the paired visible burrow entrance. And then, we continued to identify whether these soil patches are bare. If bare, we divided them into soil bare patches caused by pika; if not bare, we did not divide them into soil bare patches caused by plateau pikas. We supplemented this information into 2.3 Field sampling.**

2. 'plateau pika presence' in the title is a bit smoother (and section 3.1).

**Answer: Thanks for your comment. We have revised "plateau pika presence' into "the presence of plateau pikas" in the title and section 3.1.**

3. 17: 'ecological service of' can be deleted here because it's mentioned below. Honestly the word 'ecological service' needs to only be stated once or maybe twice in the abstract; it's continued usage is redundant. Also in the main text. A previous reviewer may have recommended this but it needn't be repeated so much.

**Answer: Great comment. According to you this comment, we have only stated once in the abstract, and the other have been deleted in the revision manuscript.**

4. The abstract doesn't contain any qualitative values and it would be of greater use to the reader if it did. 'Possible pattern' is also speculative, you can say what the pikas did in your study.

**Answer: Thank you for your comment. We have revised the abstract and result sectors, in which some qualitative values are added into manuscript. This study showed that the forage available to livestock and water conservation were 19.74% and 15.86% lower in the presence of plateau pikas than in their absence, while biodiversity conservation,**

carbon sequestration, soil nitrogen, and phosphorus maintenance were 14.58%, 29.15%, 9.97% and 8.89% higher in the presence of plateau pikas than in their absence.

In addition, we have deleted possible, which confirmed that the findings of this study present a pattern of plateau pikas influencing the ecosystem services of meadow ecosystems in alpine regions.

5. 62: 'with averaging' isn't correct usage. A few minor usage changes throughout the manuscript would make for an improvement, using an automated grammar checker will probably catch almost all instances. (see also 66 'with average' and numerous other instances, e.g. line 137 'can turn green until July'. "don't turn green until July"?).

**Answer: Thank you for your suggestion. Based on your comment, we have revised "with averaging" into "with an average weight of"; We have revised "with average" into "with an average length and depth of"; we have revised "can turn green until July" into "don't turn green until July". In addition, we have used an automated grammar checker to make for an improvement throughout the manuscript.**

6. 192: use 'W' if it is the shape of a W. I wasn't sure if it meant "West".

**Answer: Thank you for your comment. We have revised "a W pattern" into " the shape of a W pattern".**

7. Figure 1: why does it go 'a' then 'b' for some bars and 'b' then 'a' for others?

**Answer: This is a statistical expression. Lower case represents a significant difference between the absence and presence of pika in Figure 1. Generally, bars with "a" represent the bigger value, and bars with "b" represent the smaller value.**

8. Fig. 2: this looks like the famous "Intermediate disturbance hypothesis" in some cases. (There's even a wikipedia page for it: https://en.wikipedia.org/wiki/Intermediate_ disturbance_hypothesis). A little bit of pika activity is good but too much and they take over. De-emphasizing the comparisons against rabbits and prairie dogs and thinking more about ecological principles could structure the Discussion more and add references, which are a bit lacking. In other words, if you expand the interesting discussion beginning line 457 a bit and focus on fundamental theory in addition to these nice examples of intermediate disturbances, the Discussion could be further improved. (This paragraph is also too long; breaking it up would help the reader.)

**Answer: Thanks for your comment. At the beginning line 457 in former revision (line 455 in present revision), we firstly focus on the Intermediate disturbance hypothesis. We looked up some literatures (Dial and Roughgarden, Ecology, 1998, 79(4), 1412-1424; Gao and Carmel, Oikos, 2020) to refer the methods that introduce the fundamental theory. The revision discussion is as follows:**

**This study also shows that the disturbance intensity of plateau pikas also affects the forage available to livestock, biodiversity conservation, water conservation, carbon sequestration, and soil total nitrogen and phosphorus maintenance. As found in plant-species richness and aboveground plant productivity (Dial and Roughgarden, 1998; Gao and Carmel, 2020), the response of plant-species richness and palatable plant biomass to the disturbance intensity of plateau pikas follow the pattern for the intermediate disturbance hypothesis in this study. In addition, the soil organic carbon stock, soil total nitrogen and phosphorus stocks at home-range scale also support the intermediate disturbance hypothesis. However, the top soil water storage does not**

conform the intermediate disturbance hypothesis.

At lower disturbance intensity, stronger competition of dominant sedges often restrains the grass to grow well (Pang and Guo, 2018) and the rare plants to coexist (Wang et al., 2012), which leads the forage available to livestock and biodiversity conservation of alpine meadows to be maintained at a low level. Although the presence of plateau pikas can increase the input of soil organic matter, this increase is low (Pang and Guo, 2017; Pang et al., 2020b), which enables the soil organic carbon sequestration and soil nitrogen and phosphorus maintenance of alpine meadows to maintain a relatively low level.

At intermediate disturbance intensity, the activities of plateau pikas improve the growth potential of grass plants (Wang et al., 2012), and increase the input of organic matter, soil total nitrogen (Li et al., 2014), organic carbon accumulation (Yu et al., 2017b), which contributes to higher the biodiversity conservation, forage available to livestock, carbon sequestration, soil total nitrogen and phosphorus maintenance services.

At higher disturbance intensity of plateau pikas, frequent bioturbation can enable all species to be at risk of going extinct (Dial and Roughgarden, 1998). Low soil water content in alpine meadows (Liu et al., 2013) only sustains the xerophytes and mesophytes, most of which are unpalatable (Pang and Guo, 2018). This contributes to relatively lower forage available to livestock and biodiversity conservation. Low vegetation biomass decreases the input resources of soil organic matter (Sun et al., 2015; Pang and Guo, 2017), contributing to a decrease in the soil organic carbon sequestration and soil nitrogen and phosphorus maintenance of alpine meadows.

Additionally, the linearly negative relationship between the water conservation of alpine meadow and disturbance intensity of plateau pikas is ascribed to evaporation and more water infiltration on bare soil patches, as the amount of water evaporation and infiltration tends to increase as the area of bare soil increases (Liu et al., 2013).

**Reviewer 5**
1. Three significant digits are probably more appropriate for values in the abstract and elsewhere; excessive precision is distracting.
**Answer: Thanks for your comment. We have used two significant digits to revise the values in abstract and results.**
2. On line 30 I'm not sure what 'richening the small mammalian herbivores' means. 'enriching our understanding'?
**Answer: Great comment. We revised the "richening the small mammalian herbivores in relation to grassland ecosystem services" into "enriching our understanding of the small mammalian herbivores in relation to grassland ecosystem services".**
3. On line 33, please note a number of relevant references in this week's Science magazine that emphasizes the importance of grassy biomes globally, https://www.science.org/journal /science
**Answer: Thank you for your comment. We have added the relevant references in this week's Science magazine that emphasizes the importance of grassy biomes globally.**
4. 238 and elsewhere: add non-breaking spaces between values and units

**Answer: Thanks for your comment. We have added non-breaking spaces between values and units throughout the manuscript.**

5. Figure 2 and elsewhere: I'm not fully convinced that a parabolic model is appropriate because it's unclear if the mechanistic response is necessarily parabolic. I can see an increase and then decrease in response to the independent variable here, although no indication that it is necessarily symmetric. A brief justification of this choice of model would be forthcoming (noting that it does align with intermediate disturbance concepts.

**Answer: Good comment. The simulation curves were not necessarily symmetric, and this is related to disturbance intensity of plateau pikas. In this field survey experiment, we could not collect all disturbance intensity of plateau (from 0 to maximum of active burrow entrances caused by plateau pikas). In the revision, we added the final regression models. The likelihood ratio test was used to select the final regression modes in Figure 2, including the palatable plant biomass (Fig. 2A), soil organic carbon stock (Fig. 2C), plant-species richness (Fig. 2D), soil total nitrogen (Fig. 2E), and phosphorus (Fig. 2F) stocks. These final regression models showed that disturbance intensities of plateau pika had a clear maximum value for the palatable plant biomass, soil organic carbon stock, plant-species richness, soil total nitrogen and phosphorus stocks, which was align with intermediate disturbance concepts. How to use the likelihood ratio test to select the final regression modes was explained in data analysis sector in original revision. We revised the relevant expression in abstract and results, and added a brief justification in the discussion.**